# Epistasis-driven identification of SLC25A51 as a regulator of human mitochondrial NAD import

Enrico Girardi [1], Gennaro Agrimi [2], Ulrich Goldmann [1], Giuseppe Fiume[1], Sabrina Lindinger[1], Vitaly Sedlyarov[1], Ismet Srndic[1], Bettina Gürtl[1], Benedikt Agerer [1], Felix Kartnig[1], Pasquale Scarcia [2], Maria Antonietta Di Noia[2], Eva Liñeiro[1], Manuele Rebsamen[1], Tabea Wiedmer [1], Andreas Bergthaler[1], Luigi Palmieri[2,3] & Giulio Superti-Furga [1,4 ✉]

About a thousand genes in the human genome encode for membrane transporters. Among these, several solute carrier proteins (SLCs), representing the largest group of transporters, are still orphan and lack functional characterization. We reasoned that assessing genetic interactions among SLCs may be an efficient way to obtain functional information allowing their deorphanization. Here we describe a network of strong genetic interactions indicating a contribution to mitochondrial respiration and redox metabolism for SLC25A51/MCART1, an uncharacterized member of the SLC25 family of transporters. Through a combination of metabolomics, genomics and genetics approaches, we demonstrate a role for SLC25A51 as enabler of mitochondrial import of NAD, showcasing the potential of genetic interaction-driven functional gene deorphanization.

[1] CeMM Research Center for Molecular Medicine of the Austrian Academy of Sciences, Vienna, Austria. [2] Laboratory of Biochemistry and Molecular Biology, Department of Biosciences, Biotechnologies and Biopharmaceutics, University of Bari, Bari, Italy. [3] CNR Institute of Biomembranes, Bioenergetics and Molecular Biotechnologies (IBIOM), Bari, Italy. [4] Center for Physiology and Pharmacology, Medical University of Vienna, Vienna, Austria. ✉email: gsuperti@cemm.oeaw.ac.at

Transmembrane transporters are key contributors to the energetic and metabolic needs of a cell. The largest human family of transporters is composed of Solute Carriers (SLCs), a diverse set of transmembrane proteins consisting of more than 400 members[1]. SLCs can be found on both the plasma membrane and in intracellular organelles, where they control the uptake and release of all major classes of biologically active molecules. Although several prominent members of the family have been the focus of extensive research, a large proportion of them remains uncharacterized[2,3], making this one of the most asymmetrically studied human protein families[4].

Genetic interactions offer a powerful way to infer gene function by a guilt-by-association principle[5,6], with negatively interacting pairs often sharing functional overlap. Positive interactions, on the other hand, can reflect regulatory connections or participation in the same protein complex. Fundamental studies in model organisms have systematically expanded our understanding of genetic regulatory networks and functionally annotated orphan genes[6,7] and recent technological developments, in particular the development of Clustered Regularly Interspaced Short Palindromic Repeats (CRISPR)-based approaches, made now possible to tackle systematic mapping of genetic interaction landscapes in human cells[8–12].

To gain insight into the interplay and functional redundancy among SLCs, we have been systematically characterizing their genetic interaction landscape in the human cell line HAP1[13]. This was achieved by combining a panel of isogenic cell lines each lacking 1 of 141 highly expressed, non-essential SLCs[14,15] with a CRISPR/Cas9 library targeting 390 human *SLC* genes[16]. Genetic interactions were calculated by comparing the cellular fitness, approximated by the corresponding single guide RNAs depletion in wild-type (wt) or SLC-deficient cells, of single and double knockout (KO) combinations[13]. The resulting dataset, comprising more than 55,000 SLC–SLC combinations at multiple time points, provides a valuable resource for the systematic generation of testable hypotheses for orphan SLCs, as exemplified below.

Approximately 60 SLCs are expressed in the mitochondria[3]. Most of these are members of the SLC25 subfamily, which currently includes 53 transporters[17,18]. Although several mitochondrial transporters remain orphans, our growing understanding of cellular and mitochondrial metabolism leads us to postulate the existence of yet uncharacterized transporters for specific metabolites[19]. Among these, the presence and nature of a mammalian mitochondrial nicotinamide adenine dinucleotide, NAD(H), carrier has been controversial[20]. NAD(H) controls cellular redox state and is a required cofactor for key metabolic pathways such as glycolysis, the tricarboxylic acid (TCA) cycle and one carbon metabolism. Moreover, NAD(H) acts as a substrate for non-redox reactions including deacetylation by sirtuins and post translation modifications by poly(ADP-ribose) polymerases[20,21]. Previous studies suggested that NAD(H) might be synthetized in the mitochondria from precursors such as nicotinamide[22] or nicotinamide mononucleotide (NMN)[23]. However, recent studies demonstrated that mitochondria can import NAD+[24], and that the cytosolic NAD(H) levels affect the mitochondrial pool[25], suggesting the presence of a transporter responsible for NAD+ compartmentalization[20]. Moreover, mitochondrial NAD+ transporters have been identified in yeast and plants[26,27] but no functional ortholog has been yet identified in humans[28]. Here we show, using an epistasis-driven approach, that the orphan solute carrier SLC25A51 controls the levels of mitochondrial NAD(H) and is the functional ortholog of the yeast mitochondrial NAD+ transporter, therefore identifying it, to the best of our knowledge, as a bona fide human mitochondrial NAD+ transporter.

## Results

**A genetic interaction network for the orphan gene *SLC25A51* suggests a role in redox and one carbon metabolism.** Within the SLC genetic interaction dataset (Fig. 1a)[13], the strongest interactions involved a small network of transporters centered around the orphan gene *SLC25A51*, encoding for Mitochondrial Carrier Triple Repeat 1 (MCART1, Fig. 1b). We observed a strong

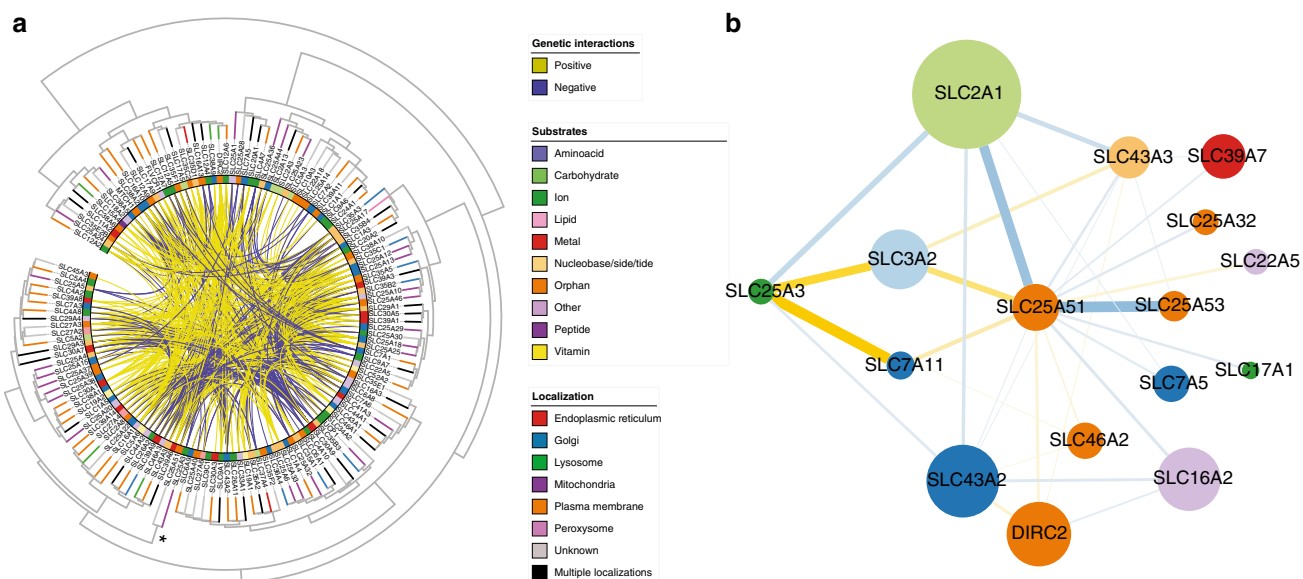

**Fig. 1 A genetic interaction network functionally annotates the orphan gene SLC25A51. a** Overview of the genetic interaction landscape of Solute Carriers in human HAP1 cells. Genes present in the SLC KO collections are arranged on a circle and clustered by genetic interaction profile similarity. Direct gene–gene interactions are shown as connections within the circle. Substrate classes are annotated in the inner color band. Localization is shown on the dendrogram leaves. *The position of the cluster composed by the mitochondrial *SLC25A3* and *SLC25A51* genes. **b** Sub-network of strong genetic interactions involving the orphan gene *SLC25A51*. Node color refers to substrate class and node size reflects the degree of connectivity in the strong SLC–SLC gene interaction network.

negative interaction of *SLC25A51* with the main glucose transporter expressed at the plasma membrane, *SLC2A1*/GLUT1, a major regulator of glycolytic metabolism[29], suggesting a potential role of SLC25A51 in cellular energetics. Moreover, we observed negative interactions with the methionine transporter genes *SLC7A5* and *SLC43A2*, as well as the paralog *SLC43A3*, which has been reported to transport nucleosides and play a role in the purine salvage pathway[30]. These connections, together with the strong positive interactions observed with *SLC7A11* and *SLC3A2*, encoding the subunits of the heterodimeric glutamate/cystine transporter xCT[31] and an important regulator of the methionine cycle and of glutathione production, suggested a potential link of SLC25A51 with one carbon and redox metabolism.

SLC25A51 is a member of the SLC25 family of mitochondrial carriers, showing close homology to the paralogs SLC25A52 and, to a lesser extent, SLC25A53[32]. Similar to the other family members, SLC25A51 is predicted to contain six transmembrane regions, with its N and C termini exposed towards the mitochondrial intermembrane space (Supplementary Fig. 1a)[33]. Functional genomics studies showed that loss of SLC25A51 can have a strong effect on cell growth[34], although it is not strictly essential in HAP1 cells[15,16]. Interestingly, and consistent with its effect on cellular fitness, <1 in 2000 individuals carry a putative deleterious mutation in SLC25A51, suggesting it is one of the most functionally conserved SLCs across humans[35]. Analysis of transcriptomics data across healthy tissues further showed that this transporter, but not its close paralog SLC25A52, is widely and robustly expressed (Supplementary Fig. 1b)[36], consistent with SLC25A51 playing an important function conserved across tissues.

Interestingly, we further observed a similar set of genetic interactions for *SLC25A51* and *SLC25A3* (Fig. 1b), a phosphate mitochondrial transporter for which a key role in oxidative phosphorylation, calcium handling, copper homeostasis, and biogenesis of complex IV of the electron transport chain has been reported[37,38], consistent with their similar genetic interaction profiles across the full dataset (Fig. 1a). Both genes showed negative interactions with *SLC2A1* and *SLC43A2*, as well as positive interactions with *SLC7A11* and *SLC3A2*, suggesting the loss of these two genes had similar effects on glycolysis and one carbon metabolism. Importantly, we further observed *SLC25A51*-specific negative interactions with two additional members of the SLC25 family, the paralog *SLC25A53* and the gene for the putative folate/FAD transporter *SLC25A32* (Fig. 1b)[39]. Although no function has been assigned to SLC25A53 yet, the interaction to SLC25A32 pointed to a possible involvement of SLC25A51 in mitochondrial cofactor transport. Overall, analysis of SLC–SLC genetic interactions resulted in the identification of cellular processes associated with loss-of-function of the broadly expressed SLC25A51 transporter, including one carbon and redox metabolism.

**SLC25A51-deficient cells have impaired mitochondrial respiration**. Prompted by the observed genetic profile similarity between the mitochondrial phosphate carrier SLC25A3 and SLC25A51, and the predicted effects on cellular energetics, we tested whether *SLC25A51*-deficient cells showed a defect in mitochondrial respiration. When measuring Oxygen Consumption Rate (OCR), SLC25A51-deficient cells, but not cells lacking the glutamate/aspartate carrier SLC25A13, showed a loss of mitochondrial respiration comparable to SLC25A3-deficient cells (Fig. 2a). Ectopic expression of SLC25A51 confirmed its predicted localization at the mitochondria, as assessed by co-staining with the organelle marker apoptosis-inducing factor (AIF, Supplementary Fig. 2a), similar to the well-characterized mitochondrial

transporter SLC25A3. Importantly, expression of *SLC25A51*, but not *SLC25A3*, restored mitochondrial respiration, suggesting that the two transporters affected OCR in non-redundant ways, likely through the transport of different substrates (Fig. 2a, b).

**Co-essentiality analysis suggests a role for SLC25A51 in mitochondrial energetics**. If indeed *SLC25A51* and *SLC25A3* encoded for complementary, non-redundant functions, then they should display correlations in a genetic co-dependency analysis[40], as genes with related functions will have similar essentiality profiles across cell lines. Consistent with our genetic interaction and OCR results, a co-dependency analysis based on the profile of essentiality of *SLC25A51* across the DepMap dataset[41] revealed *SLC25A3* as the most correlated SLC, as well as high correlation to genes encoding subunits of the electron transport chain, ATP synthase and the mitochondrial ribosome (Fig. 3a), supporting a possible role of SLC25A51 in enabling mitochondrial respiration. Among the other, highly correlating SLCs, we observed several mitochondrial cofactor transporters, including the SAM transporter SLC25A26[42], the thiamine transporter SLC25A19[43], and the folate/FAD transporter SLC25A32 (Fig. 3a). Interestingly, analysis of the co-dependencies specific to *SLC25A51* and not *SLC25A3* highlighted a similarity to mutations of pyruvate dehydrogenase and citrate synthase (Supplementary Fig. 2b), suggesting that loss of SLC25A51 mimicked the loss of enzymes acting at the early stages of the TCA cycle, several of which rely on NAD, thiamine, and CoA for their function. Overall, co-essentiality analysis further supported an involvement of SLC25A51 in mitochondrial energetics, possibly through transport of a key cofactor.

**Comparative whole-cell and mitochondrial targeted metabolomics identifies key metabolites affected by SLC25A51 loss**. To test whether loss of SLC25A51 function was consistent with the functional annotation proposed by its SLC genetic interaction network and the co-essentiality analysis, we characterized the metabolic changes in cells lacking either SLC25A51 or SLC25A3 compared to wt cells by measuring the abundances of a panel of 194 metabolites by liquid chromatography–tandem mass spectrometry (LC-MS/MS) (Fig. 4a–c and Supplementary Fig. 2c). We observed a similar pattern of changes, including depletion of riboflavin, purine nucleotides AMP and IMP, and enrichment of their precursor AICAR (5-aminoimidazole-4-carboxamide ribonucleotide), as well as increased levels of metabolites associated to glutathione metabolism, pointing to a defect in one carbon and redox metabolism (Fig. 4b). This is consistent with the previously mentioned negative interactions of *SLC25A51* with methionine transporters, the putative mitochondrial folate/FAD transporter *SLC25A32*, as well as the purine transporter *SLC43A3* (Fig. 1b). Interestingly, comparison of changes in metabolites between SLC25A51- and SLC25A3-deficient cells showed stronger depletion of TCA cycle intermediates, including citrate, aconitate and cis-aconitate, isocitrate and succinate, in SLC25A51 knockouts (Fig. 4d), suggesting that comparison of metabolic profiles from whole cells lacking functionally related genes can provide insights into specifically affected processes consistent with genetic analyses.

Based on the evidence so far accumulated, we reasoned that SLC25A51 could provide a small molecule or cofactor involved with both the early stages of the TCA cycle as well as the assembly/functioning of the electron transport chain (ETC) (Fig. 5a). Among these cofactors and vitamins, NAD was the only significantly depleted molecule in SLC25A51-deficient cells vs. SLC25A3-deficient ones (Fig. 4a–d), leading us to hypothesize that SLC25A51 could be involved in the uptake of NAD(H) or its

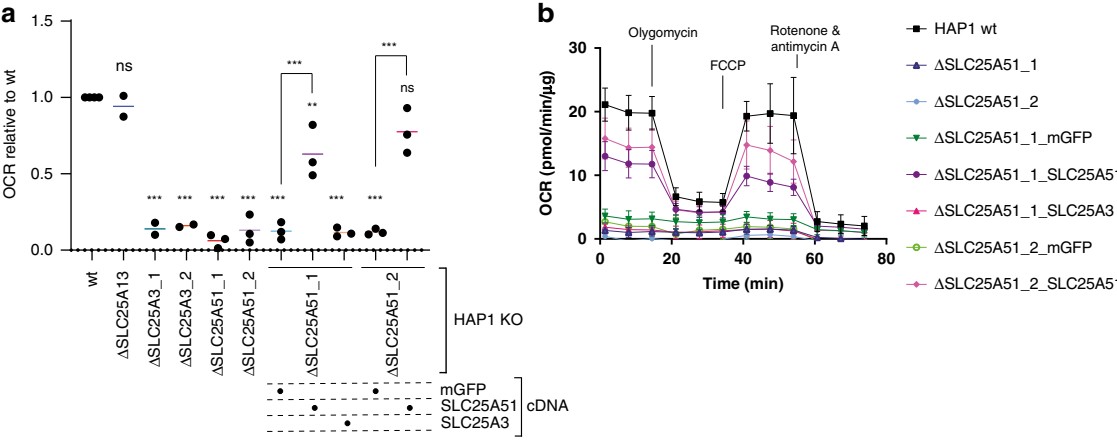

**Fig. 2 ΔSLC25A51 cells have reduced mitochondrial respiration. a** Oxygen consumption rate (OCR) in wt and SLC-deficient HAP1 cells, as well as cells reconstituted with the indicated cDNAs. mGFP: mitochondrially localized GFP. SLC25A13 KO cells are used as negative control ($n = 2$–$3$ independent biological replicates). The bars show average value across biological replicates. Statistical significance was calculated with a one-way ANOVA with Dunnett's multiple comparison test. ns: nonsignificant, **$p$-value < 0.05, ***$p$-value < 0.01. Source data are provided as a Source Data file. **b** Oxygen consumption rate from a representative Seahorse measurement with ΔSLC25A51 cells. Average and SD of five to eight technical replicates ($n = 1$ shown as representative experiment). Source data are provided as a Source Data file. FCCP: carbonyl cyanide 4-(trifluoromethoxy)phenylhydrazone.

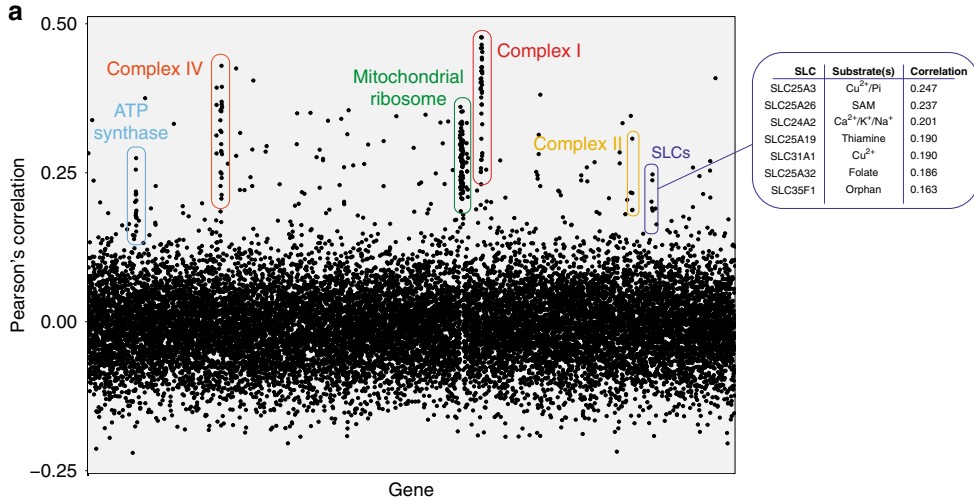

**Fig. 3 A co-essentiality analysis functionally links SLC25A51 to the ETC and cofactor transport. a** Plot of the correlations of essentiality across cell lines in the DepMap dataset for each human gene, in relation to SLC25A51. Members of major mitochondrial complexes and SLC with the most highly correlated essentiality profiles are labeled.

precursors. Consistent with these results, NAD(H) levels were reduced in SLC25A51-deficient cells compared to SLC25A3 knockouts, when measured by a luminescence-based assay (Supplementary Fig. 2c). To further investigate this possibility, we measured metabolite abundance in samples enriched in mitochondria from wt or SLC25A51-deficient cells (Fig. 5b and Supplementary Fig. 2d). NAD was the most depleted metabolite in SLC25A51-deficient mitochondria, with several other TCA metabolites also being reduced. To further establish the effect of SLC25A51 on mitochondrial NAD levels, we measured NAD content in mitochondria from cells lacking SLC25A51, or KOs reconstituted with either mitoGFP, SLC25A51, or SLC25A3. Consistent with a role for SLC25A51 in NAD transport, this nucleotide was drastically depleted in ΔSLC25A51 cells, compared to wt (Fig. 5c). Importantly, overexpression of SLC25A51, but not mitoGFP or SLC25A3, restored the levels of NAD within the mitochondria (Fig. 5c). These findings therefore suggest that SLC25A51 affects multiple metabolic pathways by controlling the levels of mitochondrial NAD(H).

**Complementation with the yeast NAD+ transporter restores mitochondrial respiration in SLC25A51-deficient cells**. We then reasoned that, if *SLC25A51* encoded a NAD(H) transporter function in human cells, the well-characterized *Saccharomyces cerevisiae* mitochondrial NAD+ transporters, Ndt1p and Ndt2p[26], should be able to functionally rescue the respiration defect and reduced NAD levels observed in ΔSLC25A51 cells. Ectopic expression of *NDT2* did not result in discernible expression in HAP1 ΔSLC25A51 cells, possibly reflecting its lower stability[26] (Supplementary Fig. 1e). However, ectopic expression of yeast *NDT1* resulted in a clear mitochondrial localization of the construct in SLC25A51-deficient cells and reversed the mitochondrial respiration defect of the SLC25A51-deficient cells (Fig. 6a, b and Supplementary Fig. 1e).

Accordingly, expression of human SLC25A51 in the Δndt1Δndt2 *S. cerevisiae* strain almost completely rescued the growth defect of the mutant in a minimal medium supplemented with ethanol and partially restored the levels of mitochondrial NAD+ (Fig. 6c, d). In summary, our results show that SLC25A51

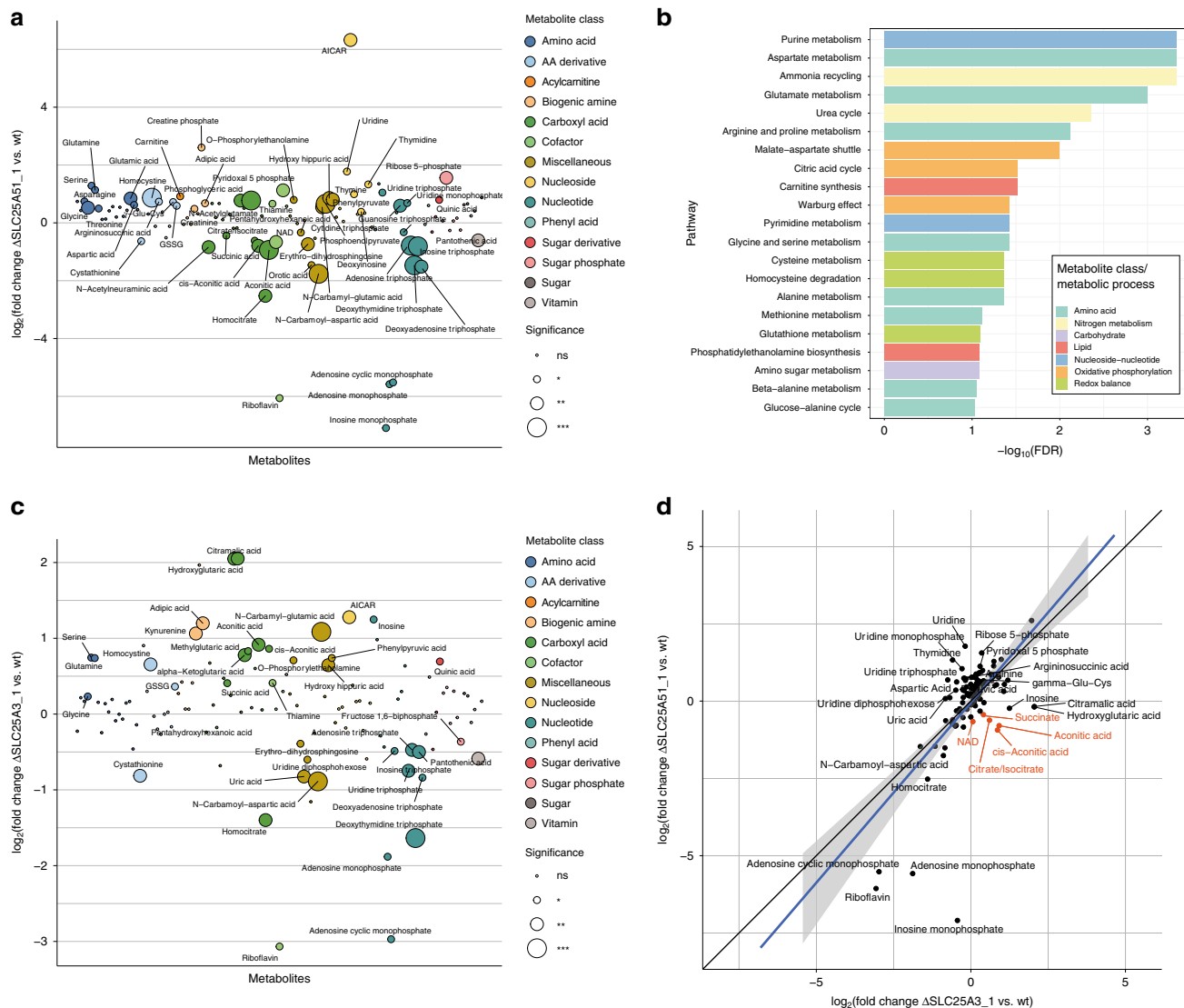

**Fig. 4 Targeted metabolomics identifies SLC25A51-specific perturbations of the TCA cycle. a** Targeted metabolomics profile of SLC25A51-deficient cells compared to wt HAP1 cells. Metabolite classes are indicated by different colors. Circle sizes reflect significance of the log$_2$ fold change measured (***p-value < 0.01, **p-value < 0.05, *p-value < 0.1, ns nonsignificant). Source data are provided as a Source Data file. **b** Enrichment analysis of metabolic pathways affected in SLC25A51 KO cells compared to wt cells, using the SMPBD database as reference. **c** Targeted metabolomics profile of SLC25A3-deficient cells compared to wt HAP1 cells. Source data are provided as a Source Data file. **d** Comparison of the log$_2$(fold change) of metabolite amounts in SLC25A51- and SLC25A3-deficient cells compared to wt. Diagonal line $y = x$ is shown in black, linear fit to the data is shown in blue with gray shaded area corresponding to 95% confidence interval. Metabolites differentially affected in the two Kos are labeled, with the subset involved with TCA cycles labeled in orange.

is the functional homolog of the well-characterized yeast mitochondrial NAD+ transporter Ndt1p.

## Discussion
The question of what is the main source of mammalian mitochondrial NAD(H), a key molecule in several biochemical and signaling pathways, has been a matter of debate, with several reports suggesting NAD+ is synthetized within the organelle from precursors such as NMN or nicotinamide due to its inability to cross the mitochondrial membrane[22,23]. In contrast, recent studies provided evidence suggesting that NAD+ is actively transported into mitochondria[24], consistent with the observed interplay between its cytoplasmic and mitochondrial pools[25]. Despite the presence of mitochondrial NAD+ carriers in yeast and plants, the nature of the transporter involved has however remained elusive[20,26,28], as none of their closest structural

homologs have been demonstrated to affect mitochondrial NAD + levels[28]. Here, starting from a network of genetic interactions among SLCs pinpointing a role for this gene in redox and one carbon metabolism, we show that loss of SLC25A51 dramatically impairs mitochondrial respiration and mimics the loss of activity of enzymes of the early steps of the TCA cycle. Through comparative targeted metabolomics approaches, we observed perturbed levels of TCA cycle metabolites and NAD(H) in whole-cell lysates from SLC25A51-deficient cells and observed drastically reduced levels of NAD(H) in mitochondria from the same cells. Finally, we showed that the yeast NAD+ transporter Ndtp1 and SLC25A51 can reciprocally complement their defective mitochondrial NAD+ levels, leading us to suggest that the previously orphan transporter SLC25A51 is required for NAD+ uptake in human mitochondria. While additional data, including transport assays with reconstituted recombinant protein, will be necessary

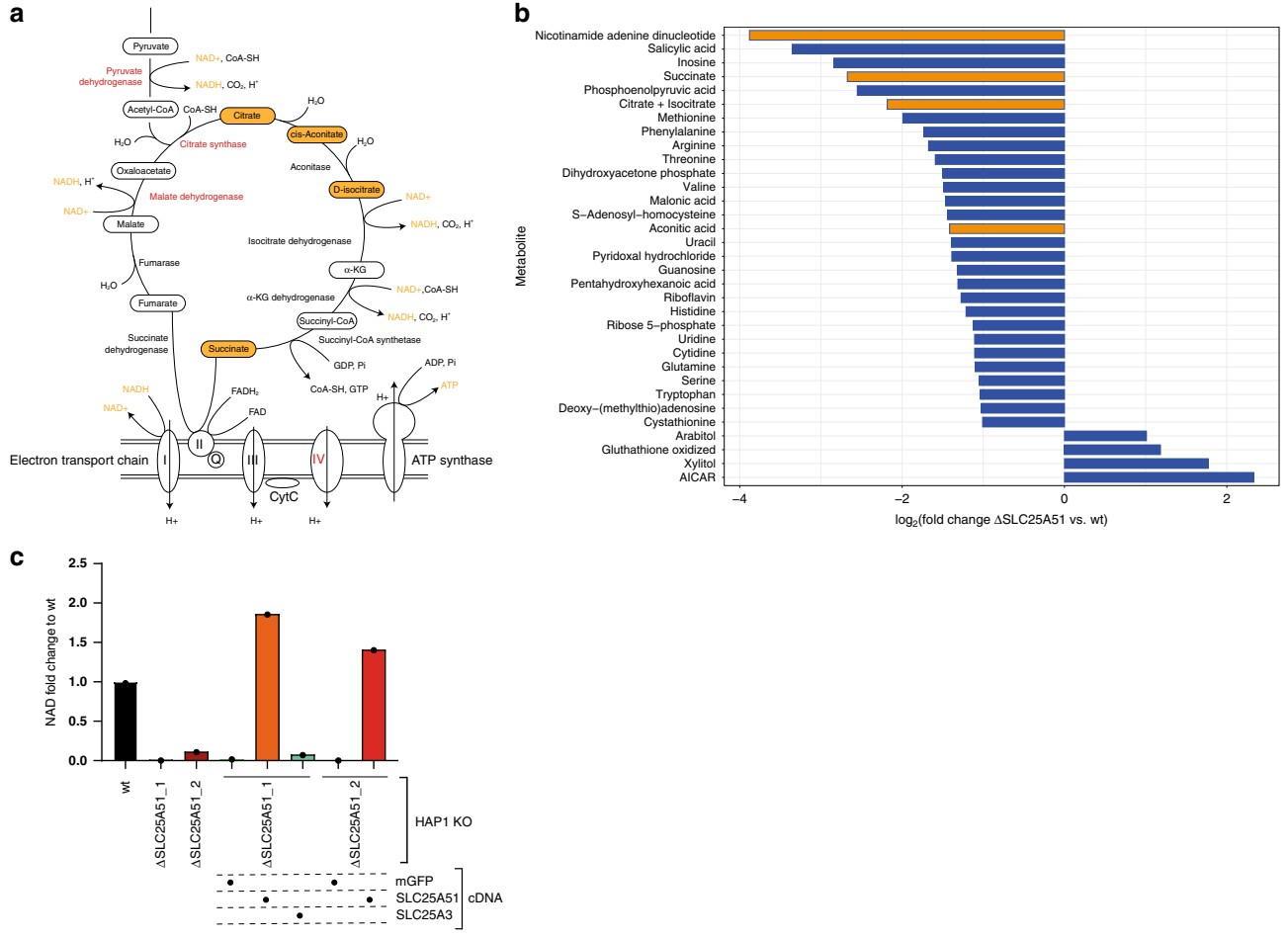

**Fig. 5 SLC25A51 controls NAD mitochondrial levels. a** Schematic view of the TCA and ETC pathway/complexes in the mitochondria. Metabolites depleted in SLC25A51 but not in SLC25A3 KOs are shown in orange boxes. Enzymes with similar essentiality profiles as SLC25A51 across the DepMap dataset are shown in red. **b** Plot of metabolites differentially abundant in ΔSLC25A51 cells (as average abundance of the two KO clones available) vs. wt cells. Only metabolites with an absolute log2(fold change) above 1 are shown (n = 2 technical replicates). Metabolites directly involved in the TCA cycle are labeled in orange. Source data are provided as a Source Data file. **c** Fold change in mitochondrial NAD content in HAP1 cells with the indicated genotypes (mean ± SD, n = 2 technical replicates). Source data are provided as a Source Data file.

to definitely prove the role of SLC25A51 as NAD+ transporter and fully define its substrate profile and transport mechanism, the multiple functional links here described convincingly converge on a functional annotation of SLC25A51 as an enabler of NAD+ accumulation in mitochondria. This annotation is corroborated by two recent publications[44,45] providing complementary evidence, including metabolomics analyses and yeast complementation experiments, of SLC25A51 being the long-hypothesized mitochondrial NAD+ transporter in human cells. Further studies addressing the interplay between SLC25A51 and the close paralogs SLC25A52, which also showed the ability to transport NAD+[44,45], and the orphan SLC25A53 are also expected to shed light on the physiological relevance and regulatory mechanisms of this SLC.

Despite the high degree of functional redundancy displayed by transporters[46], we show here that informative patterns of genetic interactions can be systematically derived for this family, allowing the generation of testable hypotheses for the de-orphanization of poorly characterized SLCs. Of all approaches that we have used in the past, including proteomics[47,48], drug resistance[16,49], and cell biological readouts[50,51], genetic interactions among SLC has the potential to be one of the most powerful, as data points contribute to construct the evidence. As in a puzzle with a limited set of pieces, if all components are evaluated, individual genes may become implicated by exclusion principles. The most severe limitation of this approach is the relatively small dimension of phenotypic space interrogated in our study: cellular fitness under a given set of growth conditions. However, in principle, the approach can be expanded to additional dimensions, by changing carbon source, or using a different read-out, such as expression of a marker gene or resistance to a specific stress. The focus on only SLC transporters, in contrast to genome-wide screens or all membrane proteins, represents at the same time a clear limitation but also a technical advantage. Genetic interactions of a larger number of genes, such as all genes in the human genome, still represents a formidable technical challenge as the number of cells analyzed and the depth of the sequencing required to obtain statistically meaningful data would elude the power of even a small consortium of laboratories. However, yet more efficient sequencing strategies, reliable robotics and robotics-compatible large cell culture incubators are likely to eventually allow to test complete genome vs. genome interactions. Meanwhile, the focus proposed here appears like a practical intermediate approach.

In conclusion, we believe that the experimental evidence obtained in this study highlights the power of using genetic interactions, and particularly the comparison of genetic interaction profiles across genes, as discovery catalyst for the functional annotation of orphan genes and SLCs in particular.

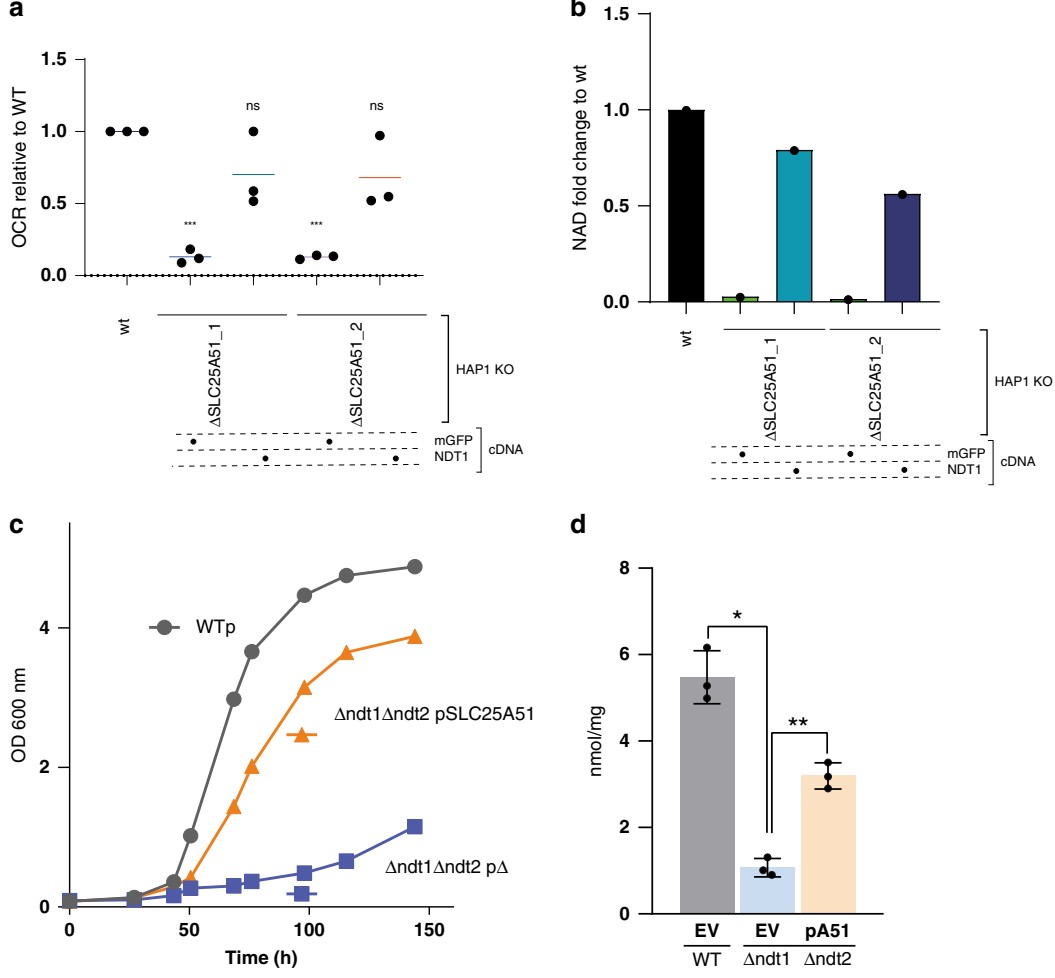

**Fig. 6 SLC25A51 is the functional ortholog of the yeast NAD+ transporter Ndt1p. a** Oxygen consumption rate (OCR) in wt and SLC25A51-deficient HAP1 cells reconstituted with mitochondrially localized GFP (mGFP) or the yeast NAD+ transporter *NTD1*. Results of three independent biological replicates are shown. The bars show average value across biological replicates. Statistical significance was calculated with a one-way ANOVA with Dunnett's multiple comparison test. ns: nonsignificant, ***p-value < 0.01. Source data are provided as a Source Data file. **b** Fold change in mitochondrial NAD content in wt or SLC25A51-deficience cells expressing mitoGFP or Ndt1p (mean ± SD, $n = 2$ technical replicates). Source data are provided as a Source Data file. **c** Effect of SLC25A51 on the growth of the Δndt1Δndt2 yeast strain. Yeast strains were grown in a YNB medium supplemented with 2% ethanol. The values of optical density at 600 nm refer to cell cultures after the indicated periods of growth. Data from a representative experiment are reported. Similar results were obtained in three independent experiments. **d** Effect of SLC25A51 on the mitochondrial NAD+ content of the Δndt1Δndt2 yeast strain. Mitochondrial extracts were assayed for NAD+ content by HPLC analysis. Data represent means ± SE of three independent experiments and were subjected to one-way analysis of variance followed by Tukey's post hoc tests (**p-value < 0.05, *p-value < 0.1). EV: empty vector, pA51: SLC25A51 plasmid.

## Methods

**Cell lines and reagents**. HAP1 cells (Haplogen Genomics) were grown in Iscove's modified Dulbecco's medium with 10% fetal bovine serum (FBS), 1% Pen/Strep. HEK293T (American Type Culture Collection) were grown in Dulbecco's modified Eagle's medium with 10% FBS, 1% Pen/Strep (Gibco). The SLC-deficient clones (ΔSLC25A51_1927-10, ΔSLC25A51_1927-11, ΔSLC25A3_792_1, ΔSLC25A3_792_6, and ΔSLC25A13_789_6, renamed as ΔSLC25A51_1, ΔSLC25A51_2, ΔSLC25A3_1, ΔSLC25A3_2, and ΔSLC25A13_1, respectively) were obtained from Haplogen Genomics. Codon-optimized SLC25A51, SLC25A3 cDNAs, or mitoGFP cDNA sequences (the latter carrying the mitochondrial import sequence derived from subunit 8 of complex IV) were obtained from the ReSOLUTE consortium (https://re-solute.eu/). The yeast NAD+ transporters ndt1/yil006w and ndt2/yel006w were obtained from Horizon Discovery (catalog IDs YSC3867-202327426 and YSC3867-202326304, respectively). cDNAs were cloned into a modified pCW57.1 lentiviral vector (Addgene #41393) generated within the ReSOLUTE consortium and carrying a Strep-HA tag and blasticidin resistance.

**Mitostress measurements**. Cells carrying inducible constructs were treated with 1 μg/ml doxycycline for 24 h before plating. To limit effects due to different doubling times across the cells lines tested, cells were seeded in 96-well plates at 40,000 cells/well on the same day of the experiment. Before measurement, medium was exchanged for XF Base Medium (Agilent 102353-100) containing glucose (10 mM), sodium pyruvate (1 mM), and L-glutamine (2 mM), and cells were incubated for 1 h at 37 °C. Measurements were carried out on a Seahorse XF96 (Agilent) with a MitoStress (Agilent, 103015-100) kit, following the manufacturer's instructions. Oligomycin, carbonyl cyanide 4-(trifluoromethoxy)phenylhydrazone, and a mix of Rotenone and Antimycin A were injected at desired time points at a final concentration of 1, 1.5, and 0.6 μM, respectively. After measurement, the medium was removed, cells were lysed, and protein amount was determined by a Bradford Assay. Data were normalized to protein amount and analyzed with Seahorse Wave 2.6.1 (Agilent) and Prism v8 (Graph Pad).

**Co-essentiality analysis**. Essentiality data were obtained from DepMap (version 19Q3). Pearson's correlations were calculated between the essentiality profiles (determined by the Ceres score[52]) of each gene across the cell line panel tested.

**Targeted metabolomics**. Cells were plated at $0.8–0.9 \times 10^6$ cells/well in six-well plates in full media. After 24 h, the cells were gently washed with room temperature phosphate-buffered saline (PBS), transferred to ice and 1.5 ml of ice-cold 80:20 MeOH:$H_2O$ solution was added to each well. The cells were scraped and transferred to a pre-cooled Eppendorf tube, snap-frozen in liquid nitrogen and thawed in ice before being centrifuged at $16,000 \times g$ for 10 min at 4 °C. Cell extracts were

dried downs using a nitrogen evaporator. The dry residue was reconstituted in 50 μL of water. Ten microliters of sample extract was mixed with 10 μL of isotopically labeled internal standard mixture in high-performance liquid chromatography vial and used for LC-MS/MS analysis. A 1290 Infinity II UHPLC system (Agilent Technologies) coupled with a 6470 triple quadrupole mass spectrometer (Agilent Technologies) was used for the LC-MS/MS analysis. The chromatographic separation for samples was carried out on a ZORBAX RRHD Extend-C18, 2.1 × 150 mm, 1.8 μm analytical column (Agilent Technologies). The column was maintained at a temperature of 40 °C and 4 μL of sample was injected per run. The mobile phase A was 3% methanol (v/v), 10 mM tributylamine, 15 mM acetic acid in water, and mobile phase B was 10 mM tributylamine, 15 mM acetic acid in methanol. The gradient elution with a flow rate 0.25 mL/min was performed for a total time of 24 min. Afterward, a back flushing the column using a 6port/2-position divert valve was carried out for 8 min using acetonitrile, followed by 8 min of column equilibration with 100% mobile phase A. The triple quadrupole mass spectrometer was operated in an electrospray ionization negative mode, spray voltage 2 kV, gas temperature 150 °C, gas flow 1.3 L/min, nebulizer 45 psi, sheath gas temperature 325 °C, and sheath gas flow 12 L/min. The metabolites of interest were detected using a dynamic Multiple Reaction Monitoring (MRM) mode. The MassHunter 10.0 software (Agilent Technologies) was used for the data processing. Ten-point linear calibration curves with internal standardization was constructed for the quantification of metabolites. Conditions were compared using Welch's t-test, p-value was subsequently corrected for multiple testing according to the Benjamini and Hochberg procedure[39]. Pathway enrichment was performed testing significantly affected metabolites ($p < 0.05$) against the Small Molecule Pathway Database[53] with MetaboAnalyst[54].

**Mitochondrial enrichment**. Subcellular fractionation of mitochondria was performed with a protocol derived from ref. [55]. Cells were homogenized in 2 ml of ice-cold mitochondrial enrichment buffer (200 mM Sucrose, 1 mM EDTA-Tris, 10 mMTris-MOPS pH 7.4) by using a Branson sonicator (30 s, Amplitude 10%, 05 s on/0.5 s off). The nuclear fraction was removed by centrifugation at $600 \times g$ for 10 min at 4 °C. Mitochondrial fraction were pelleted from the supernatant by centrifugation at $10,000 \times g$ for 15 min at 4 °C. The final mitochondrial pellet were subsequently washed with 500 μl of mitochondrial enrichment buffer ($10,000 \times g$ for 15 min at 4 °C) before being subjected to MeOH:H$_2$O extraction and LC/MS analysis. Mitochondrial enrichment was confirmed by western blotting using antibodies against alpha-tubulin (DM1A, Abcam #7291, 1:5000) and VDAC (Abcam #15895, 1:1000).

**Yeast experiments**. The yeast strains used in all experiments were the wt BY4742 (MATα his3Δ1 leu2Δ0 lys2Δ0 ura3Δ0) and the isogenic Δndt1Δndt2 (BY4742 ndt1::KanMX ndt2::HphMX)[26]. The SLC25A51 CDS was cloned in the HindIII/BamHI restriction sites of the yeast expression vector pYES2 in which the inducible GAL1 promoter had been replaced with the constitutive TDH3 promoter and carried a V5-tag at its C terminus. For the initial propagation, yeast cells were grown in rich (YP) medium, containing 2% Bacto-peptone and 1% yeast extract, supplemented with 2% glucose. Growth complementation assays were carried out in YNB medium pH 4.5 supplemented with 2% EtOH and the appropriate auxotrophic nutrients. Mitochondria were isolated from cells grown in ethanol-supplemented YP until the early exponential phase (optical density between 1.0 and 1.5) was reached[26]. For the NAD+ HPLC analysis, nucleotides were extracted from mitochondria by addition of 1.2 M ice-cold perchloric acid followed by neutralization with 30% KOH. After centrifugation and filtration, supernatants were subjected to HPLC analysis using a Waters Alliance 2695 separation module (Waters, Milford, MA, USA) equipped with a Kinetex EVO C18 column (Phenomenex, 150 mm × 4.6 mm, 100 Å, 5 μm), coupled to a Waters 2996 UV detector set at at 254 nm. Separation was carried out at 25 °C using two mobile phases consisting of 20 mM phosphate buffer with 8 mM tetra-n-butyl-ammonium bisulfate (TBA) pH 6 (solvent A) and 50% acetonitrile (solvent B) at a flow rate of 0.7 ml/min. Solvent A was held for the first 12 min; then a linear gradient profile started reaching 10% B in 4 min and 20% B in further 4 min; then it was increased up to 40% in the following 2 min and finally was kept constant for the last 8 min of separation. The initial conditions were then restored in 15 min.

**Confocal microscopy**. For the confocal imaging of HAP1 cells cells, high precision microscope cover glasses (Marienfeld) were coated with poly-L-lysine hydrobromide (p6282, Sigma-Aldrich) according to the manufacturer's protocol. Cells were seeded onto cover glasses in normal growth medium and fixed in 4% formaldehyde solution (AppliChem) in PBS 1× after 24 h of incubation. Permeabilization and blocking of samples was performed in blocking solution (5% fetal calf serum, 0.3% Triton X-100 in PBS 1×) for 1 h rocking. Anti-HA Tag (Roche #11867423001) and anti-AIF (CST #5318) primary antibodies were diluted 1:500 in antibody dilution buffer (1% bovine serum albumin, 0.3% Triton X-100 in PBS 1×) and applied for 2 h at room temperature, rocking. Samples were washed three times in antibody dilution buffer and anti-rat Alexa Fluor 488 (Thermo Fischer Scientific #A11006) and anti-rabbit Alexa Fluor 594 (Thermo Fischer Scientific #A11012) secondary antibodies were applied 1 : 500 in antibody dilution buffer for 1 h at room temperature, rocking. After three times washing in antibody dilution buffer,

nuclei were counterstained with 4′,6-diamidino-2-phenylindole 1:1000 in PBS 1×, for 10 min, rocking. Cover glasses were mounted onto microscopy slides using ProLong Gold (Thermo Fischer Scientific #P36934) antifade mountant. Image acquisition was performed on a confocal laser scanning microscope (Zeiss LSM 780, Carl Zeiss AG), equipped with an Airyscan detector using ZEN black 2.3 (Carl Zeiss AG).

**NAD/NADH and ATP measurements**. For luminescence-based assays, 10,000 HAP1 cells/well were plated in a 96-well plate in triplicates. After 6 h, ATP and NAD(H) levels were measured by CellTiterGlo and NAD/NADH-Glo assays (Promega). Readings were normalized to cell numbers measured with Casy (OMNI Life Sciences) on a mirror plate.

**Data analysis and visualization**. Exploratory data analysis and visualizations were performed in R-project version 3.6.0 (47) with RStudio IDE version 1.2.1578, ggplot2 (3.3.0), dplyr (0.8.5), readr (1.3.1).

**Reporting summary**. Further information on research design is available in the Nature Research Reporting Summary linked to this article.

## Data availability
Code available upon request. The DepMap and SMPDB databases are available at www.depmap.org and www.smpdb.ca, respectively. Source data are provided with this paper. Metabolomics datasets were deposited at Metabolights (https://www.ebi.ac.uk/metabolights/) with accession code MTBLS2205.

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

## Acknowledgements

We are grateful to the ProMet and Biomedical Sequencing (BSF) facilities at CeMM/MUW. We thank the members of the Superti-Furga and Menche groups for critical discussions. We acknowledge support by the Austrian Academy of Sciences, the European Research Council (ERC AdG 695214, E.G., G.F., M.R., ERC 677006, A.B.) and the Austrian Science Fund (FWF P29250-B30 VITRA, E.G., J.K., G.F.; FWF DK W1212, B.A., A.B.).

## Author contributions

E.G.: conceptualization, methodology, software, formal analysis, investigation, validation, visualization, project administration, and writing. G.A.: investigation, formal analysis, and visualization. U.G.: methodology, software, data curation, formal analysis, and visualization. G.F. and S.L.: investigation and validation. V.S.: data curation, formal analysis, and visualization. I.S., F.K., P.S., M.A.D.N., and E.L.: investigation. B.G. and B.A.: investigation and formal analysis. M.R.: conceptualization. T.W.: methodology. A.B.: formal analysis and funding acquisition. L.P.: conceptualization, project administration, and funding acquisition. G.S.-F.: conceptualization, project administration, funding acquisition, and writing.

## Competing interests

The authors declare no competing interests.
