## [Peer Review File · Nature Communications]

REVIEWERS' COMMENTS

Reviewer #2 (Remarks to the Author):

In the manuscript "Epistasis-driven identification of SLC25A51 as a regulator of human mitochondrial NAD import" Girardi et al. conclude that SLC25A51 is required for NAD accumulation into mitochondria by catalyzing the import of NAD⁺ or a related compound.

Several lines of evidence support this conclusion. SLC25A51 KO cells displayed a reduction of mitochondrial NAD(H) content and a dramatic impairment of mitochondrial respiration. This phenotype was rescued by ectopic expression of the yeast mitochondrial NAD⁺ transporter, Ndt1p. Conversely, SLC25A51 expression in a yeast strain devoid of mitochondrial NAD⁺ transporters (Ndt1p and Ndt2p), led to recovery of the mitochondrial NAD pool and rescued growth on non-fermentative carbon sources. Most importantly, the authors describe an original genetic interaction network approach, that complements several ongoing genome-wide CRISPR-Cas9-based screens in different cell lines and has the potential to drive the identification of other hitherto unidentified Solute Carriers (SLCs). Indeed, to explore the function of SLCs they mapped genetic interaction profiles by scoring cell fitness upon transduction of HAP1 cells in which one of 141 SLCs previously identified as "non-essential" had been knocked-out, with a CRISPR/Cas9 library targeting 390 SLC genes. Amongst them, SLC25A51 peaked as a major interaction hub pinpointing a role of this SLC in redox and one-carbon metabolism and prompting further investigations which included state-of-the-art cell metabolic analysis and targeted metabolomics. The experiments reported are technically sound. The manuscript is well written and of general interest. Furthermore, my concerns about a previous version of this manuscript submitted to Nature have been addressed satisfactorily. This reviewer acknowledges that the two recently published papers on the subject (Luongo et al. Nature 2020, Kory et al. Science Advances 2020) and this manuscript have been circulating in review processes of various journals this year, and all three are independent works performed at the same time. Considering the above, and the fact that the present results are not only interesting per se but they also demonstrate how epistasis can shed light into unknown SLCs functions, the manuscript is recommended for publication in Nature Communications provided that the below comments are dealt with.

Minor points

1. For the sake of clarity, a few more words on the way cell fitness was scored and how the score-threshold for identifying genetic interaction was set, would be beneficial.
2. The evidence provided unequivocally demonstrates that SLC25A51 regulates the accumulation of NAD(H) in the mitochondrial matrix. Yet, as the authors concede in the Discussion, its biochemical identification awaits direct transport assays with the purified (recombinant) protein reconstituted into artificial lipid vesicles (liposomes). Thereof, the final sentence of the Introduction where the authors claim to have identified it "as a bona fide human mitochondrial NAD⁺ transporter" is inappropriate and should be changed.
3. Even though the genetic interaction screen appears very powerful in linking orphan SLCs to the cellular and physiological context where they play a role, the approach needs further validation with other orphan SLCs before it "can be considered the most efficient". Hence the sentence at lines 237-240 should be tuned down.
4. Lines 111-122. The information in the section "SLC25A51 is a widely expressed and rarely mutated gene" should be incorporated in the Introduction and Discussion because the statements are extracted from literature and data banks and not based on results obtained in this manuscript.
5. Lines 117-119: "Interestingly, and consistent with its effect on cellular fitness, less than 1 in 2000 individuals carry a putative deleterious mutation in SLC25A51, making it one of the most functionally conserved SLCs across humans [38]". It is true that reference 38 says that in the 141 456 individuals

studied SLC25A51 has predicted loss-of-function variants very rarely compared to other SLCs. However, this does not necessarily mean that SLC25A51 is one of the most functionally conserved transporters in humans because the loss-of-function variants are predicted by bioinformatic approaches and their deleterious effects on function have not been experimentally confirmed. Therefore, the sentence at lines 117-119 should be tuned down.

6. For non-expert readers, it would be helpful to have more details in the text on co-dependency analysis.

7. In Suppl Figure 2 (or otherwise in the corresponding legend), panels c) and d) have been swapped.

Reviewer #3 (Remarks to the Author):

The authors made substantial efforts and added new results and discussion to address the questions raised by the reviewers. The revised manuscript has better focus and provides important and new information in the field of NAD and mitochondrial biology research. Although the in-depth functional assessments could further improve the manuscript, these experiments are beyond the scope of the study and future studies will address these important questions. This reviewer has no further comments.

REVIEWERS' COMMENTS

Reviewer #2 (Remarks to the Author):

In the manuscript "Epistasis-driven identification of SLC25A51 as a regulator of human mitochondrial NAD import" Girardi et al. conclude that SLC25A51 is required for NAD accumulation into mitochondria by catalyzing the import of NAD⁺ or a related compound. Several lines of evidence support this conclusion. SLC25A51 KO cells displayed a reduction of mitochondrial NAD(H) content and a dramatic impairment of mitochondrial respiration. This phenotype was rescued by ectopic expression of the yeast mitochondrial NAD⁺ transporter, Ndt1p. Conversely, SLC25A51 expression in a yeast strain devoid of mitochondrial NAD⁺ transporters (Ndt1p and Ndt2p), led to recovery of the mitochondrial NAD pool and rescued growth on non-fermentative carbon sources. Most importantly, the authors describe an original genetic interaction network approach, that complements several ongoing genome-wide CRISPR-Cas9-based screens in different cell lines and has the potential to drive the identification of other hitherto unidentified Solute Carriers (SLCs). Indeed, to explore the function of SLCs they mapped genetic interaction profiles by scoring cell fitness upon transduction of HAP1 cells in which one of 141 SLCs previously identified as "non-essential" had been knocked-out, with a CRISPR/Cas9 library targeting 390 SLC genes. Amongst them, SLC25A51 peaked as a major interaction hub pinpointing a role of this SLC in redox and one-carbon metabolism and prompting further investigations which included state-of-the-art cell metabolic analysis and targeted metabolomics. The experiments reported are technically sound. The manuscript is well written and of general interest. Furthermore, my concerns about a previous version of this manuscript submitted to Nature have been addressed satisfactorily. This reviewer acknowledges that the two recently published papers on the subject (Luongo et al. Nature 2020, Kory et al. Science Advances 2020) and this manuscript have been circulating in review processes of various journals this year, and all three are independent works performed at the same time. Considering the above, and the fact that the present results are not only interesting per se but they also demonstrate how epistasis can shed light into unknown SLCs functions, the manuscript is recommended for publication in Nature Communications provided that the below comments are dealt with.

We take the opportunity to thank the reviewer for the valuable feedback and suggestions throughout the revision process.

Minor points

1. For the sake of clarity, a few more words on the way cell fitness was scored and how the score-threshold for identifying genetic interaction was set, would be beneficial.

We thank the reviewer for the suggestion. We have now added an additional sentence to clarify how genetic interactions were calculated and cite the corresponding preprint for further details on thresholds used.

2. The evidence provided unequivocally demonstrates that SLC25A51 regulates the accumulation of NAD(H) in the mitochondrial matrix. Yet, as the authors concede in the Discussion, its biochemical identification awaits direct transport assays with the purified (recombinant) protein reconstituted into artificial lipid vesicles (liposomes). Thereof, the final sentence of the Introduction where the authors claim to have identified it "as a bona fide human mitochondrial NAD⁺ transporter" is inappropriate and should be changed.

We understand the reviewer's concerns and modified the sentence to reflect the suggestive, but not yet definitive, nature of SLC25A51 as mitochondrial NAD(H) transporter. However, we left "bona fide" as it means "in good faith". There are no real alternatives from our screen and the evidence, compiled, is compelling.

3. Even though the genetic interaction screen appears very powerful in linking orphan SLCs to the cellular and physiological context where they play a role, the approach needs further validation with other orphan SLCs before it "can be considered the most efficient". Hence the sentence at lines 237-240 should be tuned down.

We modified the sentence to reflect the current potential of genetical interactions to reveal novel functional annotations for SLCs.

4. Lines 111-122. The information in the section "SLC25A51 is a widely expressed and rarely mutated gene" should be incorporated in the Introduction and Discussion because the statements are extracted from literature and data banks and not based on results obtained in this manuscript.

We respectfully disagree and believe that this section provides important information to place SLC25A51 into context within the results of our study. To simplify the manuscript flow we have however now merged this section with the previous one.

5. Lines 117-119: "Interestingly, and consistent with its effect on cellular fitness, less than 1 in 2000 individuals carry a putative deleterious mutation in SLC25A51, making it one of the most functionally conserved SLCs across humans [38]". It is true that reference 38 says that in the 141 456 individuals studied SLC25A51 has predicted loss-of-function variants very rarely compared to other SLCs. However, this does not necessarily mean that SLC25A51 is one of the most functionally conserved transporters in humans because the loss-of-function variants are predicted by bioinformatic approaches and their deleterious effects on function have not been experimentally confirmed. Therefore, the sentence at lines 117-119 should be tuned down.

We thank the reviewer for raising this point. We modified the corresponding sentence to tone down our conclusions.

6. For non-expert readers, it would be helpful to have more details in the text on co-dependency analysis.

Many thanks for highlighting this point. We added an extra sentence to explain in more detail the co-dependency approach.

7. In Suppl Figure 2 (or otherwise in the corresponding legend), panels c) and d) have been swapped.

Now corrected in the updated version. Many thanks for pointing this to us.

Reviewer #3 (Remarks to the Author):

The authors made substantial efforts and added new results and discussion to address the questions raised by the reviewers. The revised manuscript has better focus and provides important and new information in the field of NAD and mitochondrial biology research. Although the in-depth functional assessments could further improve the manuscript, these experiments are beyond the scope of the

study and future studies will address these important questions. This reviewer has no further comments.

We are very grateful to the reviewer for his assessment and, as suggested, will continue to characterize SLC25A51 in future studies.